# Glycovariant-based lateral flow immunoassay to detect ovarian cancer–associated serum CA125

Sherif Bayoumy [1], Heidi Hyytiä [1,5], Janne Leivo[1], Sheikh M. Talha [1], Kaisa Huhtinen [2], Matti Poutanen [3], Johanna Hynninen [4], Antti Perheentupa [4], Urpo Lamminmäki[1], Kamlesh Gidwani[1] & Kim Pettersson [1]✉

Cancer antigen 125 (CA125) is a widely used biomarker in monitoring of epithelial ovarian cancer (EOC). Due to insufficient cancer specificity of CA125, its diagnostic use is severely compromised. Abnormal glycosylation of CA125 is a unique feature of ovarian cancer cells and could improve differential diagnosis of the disease. Here we describe the development of a quantitative lateral flow immunoassay (LFIA) of aberrantly glycosylated CA125 which is widely superior to the conventional CA125 immunoassay (CA125IA). With a 30 min read-out time, the LFIA showed 72% sensitivity, at 98% specificity using diagnostically challenging samples with marginally elevated CA125 (35–200 U/mL), in comparison to 16% sensitivity with the CA125IA. We envision the clinical use of the developed LFIA to be based on the substantially enhanced disease specificity against the many benign conditions confounding the diagnostic evaluation and against other cancers.

[1] Department of Biochemistry/Biotechnology, University of Turku, Turku, Finland. [2] Department of Pathology, Institute of Biomedicine, Research Center for Cancer, Infections and Immunity, University of Turku, Turku, Finland. [3] Institute of Biomedicine, Research Center for Integrative Physiology and Pharmacology and Turku Center for Disease Modeling, University of Turku, Turku, Finland. [4] Department of Obstetrics and Gynecology, Turku University Hospital, Turku, Finland. [5] Present address: PerkinElmer Finland Oy, Turku, Finland. ✉email: Kim.Pettersson@utu.fi

ateral flow immunoassay (LFIA) (i.e., immunochromatographic assay) is a well-established platform in the field of point-of-care (POC) diagnostics. The platform provides real-time results, simple operation, and good shelf life (e.g., pregnancy test kits). Different commercial LFIAs have been available for the detection of tumor protein biomarkers[1]. Nevertheless, cancer biomarker tests are almost exclusively carried out in centralized laboratories, thus restricting the POC diagnostics. Epithelial ovarian cancer (EOC) is the deadliest form of gynecological malignancies because of late and vague symptoms, frequently seen in many other conditions[2]. Another reason is the difficulty of diagnosing the localized malignancy at an early-stage using existing methods[3,4]. Transvaginal ultrasonography (TVUS) and quantitative measurement of CA125 enable detection of mostly advanced EOC[5]. CA125 commonly also rises in several benign gynecologic conditions. In cases of clinical suspicion of EOC, there is a need for a simple POC test, that doctors could conduct at time of presentation. By measuring ovarian cancer-associated CA125, the high-risk patients could be directed rapidly to a specialist for definitive diagnostic work-up. A simple CA125 test of decisively increased EOC specificity is pivotal to reduce false-positive results that expose patients to unnecessary invasive diagnostic procedures and to improve the treatment outcome and the survival rate of the patients[6].

Quantification of CA125 has been performed using conventional CA125 immunoassays (CA125IA). The CA125IA, ELISA test, utilize the specificity of different monoclonal antibodies targeting protein-epitopes on CA125, including OC125 and M11, or OV197 like antibodies[7]. These assays have been widely implemented for the monitoring of disease progression or regression, rather than for early detection of ovarian cancer. The inadequate specificity of CA125 impedes its use in early-stage EOC diagnosis and disease progression[5,8,9]. For this reason, supplementary biomarkers to CA125 such as HE4[10] or multimodal diagnostic tests (ROMA, ROCA, OVA1, and Overa) with CA125 as the key component have been studied[11]. More recently, several endeavors combined multiple tumor markers to increase clinical sensitivity and specificity of EOC detection, but CA125 has remained the preferred biomarker[12,13]. The results of a large prospective ovarian cancer screening study (UKTOCS), combining ultrasound and serial CA125 measurements, showed that this screening strategy may improve early detection and reduce disease mortality[5].

The normal range of circulatory CA125 concentration for the diagnosis and management of EOC is commonly set below 35 U/mL[6]. Therefore, levels >35 U/mL and increasing levels over time are considered a potential indication of malignancy[5,6]. CA125 is a mucin-type glycoprotein encoded by the *MUC16* gene. The gene comprises a short cytoplasmic domain, a transmembrane region and a large extracellular domain. The extracellular domain is rich in *N*- and *O*-linked glycosylation[2,14]. Glycosylation patterns between normal and malignant cells are different, because of the altered or truncated carbohydrate side chains in malignancy. Expression of the sialyl-Tn antigen (STn, Neu5Acα2, 6GalNAc *O*-Ser/Thr) is limited in normal cells[15]. However, premature sialylation of the core carbohydrate Tn structure Gal-NAc1-*O*- Ser/Thr in malignant epithelium  is reported to lead to excessive expression of the STn antigen[15,16]. The aberrant glycosylation is conjoined with malignant transformation, tumor progression, and metastasis of many tumor types and leads to altered serum glycoprofiles[17]. Notably, targeting aberrant elevated STn expression has attracted interest due to the improved clinical specificity and sensitivity for ovarian cancer detection[18,19]. Therefore, glycan biomarkers of EOC could be a viable differential diagnostic tool. Gidwani et al.[20,21] recently demonstrated a CA125 glycovariant-based assay utilizing fluorescent-europium nanoparticles, which improves discrimination of EOC from benign endometriosis disease compared to the conventional immunoassays.

The inadequate clinical performance of traditional LFIAs has restricted the implementation of the platform for the detection of tumor biomarkers[22]. Commonly, a LF strip consists of a nitrocellulose membrane, sample pad, conjugate pad, and absorbent pad, all laminated on a plastic backing card and placed in a plastic housing[23]. Currently available LFIAs are either qualitative or semi-quantitative assays with limited sensitivity for detecting challenging analytes[22]. The detection method is usually based on visual interpretation of colloidal gold or colored nanoparticle reporters. Moreover, subjective visualization of test results has always been a major concern that could lead to false results. For this reason, there is a growing need to develop quantitative and reader instrument-based LFIAs. Generally, fluorescent reporters provide superior sensitivity to colloidal gold and allow quantification of analytes[24,25]. Among fluorescent reporters, upconverting nanoparticles (UCNPs; Upcon® nanoparticles, Kaivogen Oy, Finland) are crystalline lanthanide-doped nanometersized particles. UCNPs are characterized by minimal auto-fluorescence background, resistance to photobleaching, and the ability to detect minute concentrations of biomarkers in biological samples[26,27]. These unique features, which are based on the near-infrared excitation and anti-Stokes shifted luminescence, assure a successful implementation of UCNPs in POC diagnostics. The low-cost of infrared laser light source enables the construction of an inexpensive portable reader[28]. The introduction of nanoparticles-based reporters has profoundly facilitated the development of high-sensitivity assays[20]. We measured the UCNPs emission using the Upcon® reader (Labrox Oy, Finland). In this work, we report a quantitative lateral flow immunoassay utilizing the UCNPs for detection of STn-glycosylated CA125 to differentiate EOC from benign endometriosis and healthy controls in patient-derived serum samples.

## Results

**Selection of antibodies**. The model analyte CA125, purified from ovarian cancer cell-line OVCAR3 (Fujirebio Diagnostics AB, Sweden), was used during the development and optimization of the assay. An anti-STn monoclonal antibody (STn1242-mAb, Fujirebio Diagnostics AB) was used to capture the aberrantly glycosylated CA125 protein on the test line of the lateral flow strips. An anti-CA125 mAb (4602, Oy Medix Biochemica, Finland), against the protein epitope was coated on the UCNPs to be used as a reporter (Fig. 1). The assay configuration based on the STn1242 and 4602 mAb-UCNPs showed significantly lower background and superior signal to noise (S/N) ratio in comparison to the remaining antibody combinations (Supplementary Fig. 1).

**Glycovariants-based approach enhances specificity to EOC-CA125**. We then compared the developed CA125-STn-LFIA to the conventional CA125-IA (Fujirebio Diagnostics AB, CanAg® CA125 EIA). The antibody combination from the CA125-IA was also tested in the lateral flow format using the UCNPs. The anti-CA125 mAb Ov197 was dispensed on the test line while the anti-CA125 mAb OV185-UCNP was used as a reporter. Pooled ascitic fluid from liver cirrhosis (LC) patients was used as nonmalignant source of the CA125 (LC-CA125) antigen. Next, we spiked pooled healthy women serum samples with OVCAR3-CA125 or LC-CA125 at different concentrations. As expected, the LFIA targeting CA125 protein-epitopes only (CA125-LFIA), showed overlapping signals of both LC-CA125 and OVCAR3-CA125. However, the CA125-STn-LFIA showed negligible signals from LC-derived CA125 (Supplementary Fig. 2). This finding underlines the significant potential of glycovariant-based

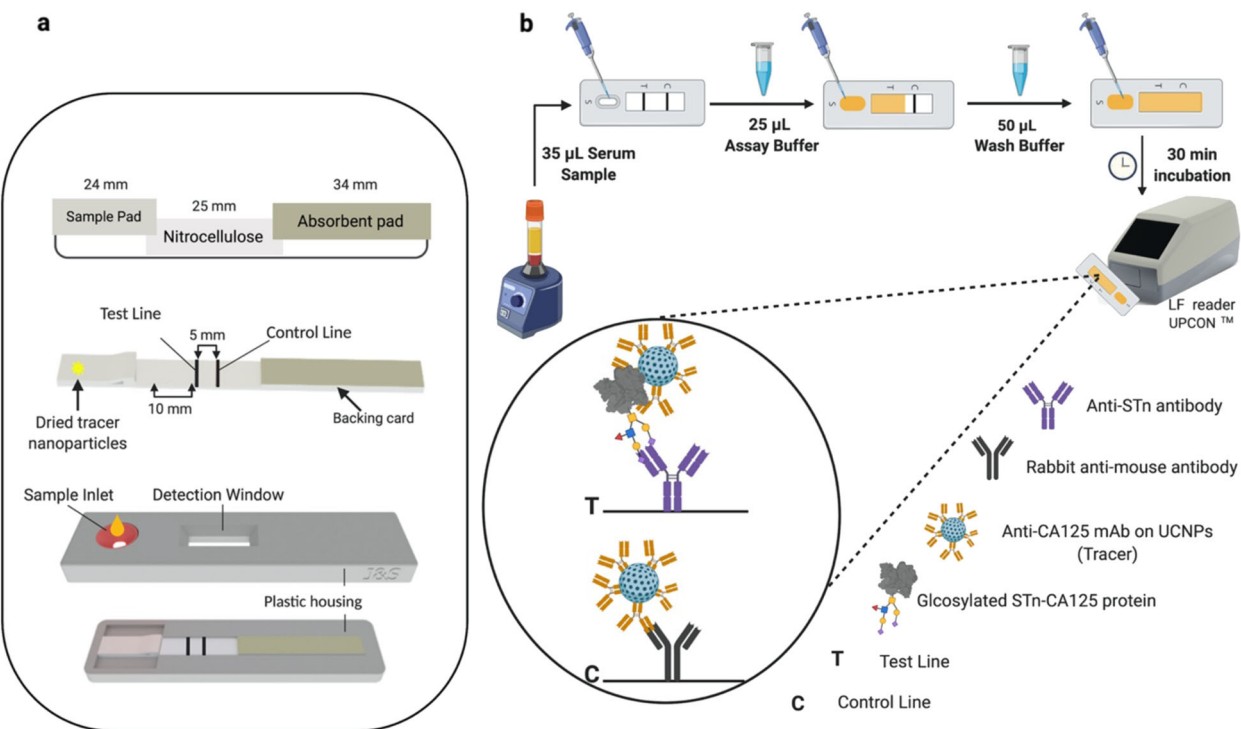

**Fig. 1 LFIA overview. a** Illustration of components of the developed LFIA strips. The developed strip is composed of a sample pad, nitrocellulose membrane, and an absorbent pad. The sample pad was used for multiple tasks, which is to promote even flow of a sample and the reporter particles to the nitrocellulose membrane. **b** Experimental workflow: protocol of the developed lateral flow immunoassay and the principle of the lateral flow immunoassay. Created with BioRender.com.

immunoassays for the discrimination of EOC from benign cases or healthy individuals (Fig. 2).

We subsequently examined the kinetics of the LFIA using the reporter 4602 mAb-UCNP at a concentration of 10 ng per strip. Three concentrations of the model analyte were spiked into serum pool of healthy individuals as calibrators, (25, 50, and 100 U/mL), and were compared to a blank calibrator (i.e., non-spiked pool of serum samples collected from healthy women). The non-specific binding clearly decreased over time until the 30 min time point. The optimum binding was reached between 30 and 40 min of incubation (Supplementary Fig. 3). In order to keep the total assay duration short, a 30 min incubation time was opted.

**Technical and clinical evaluation.** To evaluate the CA125-STn LFIA, we determined the limit of detection (LoD) using the standard curve as shown in Supplementary Fig. 4. The LFIA had a LoD of 5.4 U/mL. To challenge the assay in the most relevant clinical situation, we selected EOC and endometriosis samples with marginally elevated CA125 concentrations (between 35 and 200 U/mL). The developed CA125-STn-LFIA assay showed that, at 98% specificity, 72% sensitivity could be achieved. On the other hand, in contrast, at a 98% specificity the CA125-IA showed a sensitivity of 16% (Fig. 2). The median values obtained from the CA125-IA in healthy controls, endometriosis, and EOC samples were 10.4, 69.6, and 70 U/mL, respectively. However, with the CA125-STn-LFIA medians were 0.04, 0.2, and 17.4 U/mL, respectively. The difference in medians of EOC and endometriosis is significant, using the CA125-STn-LFIA, in comparison to the conventional CA125-IA (Fig. 2). Using the LFIA, most of the healthy and endometriosis samples showed CA125 concentration below the assay LoD. Therefore, we advocate establishing the boxplot with test line signals instead of quantification of the glycoprotein.

## Discussion

In this study, we demonstrate a quantitative glycovariant-based lateral flow immunoassay that seeks to detect EOC-associated serum CA125. The developed LFIA offers immediacy of response (30 min), simplicity, and superior clinical performance in comparison to the reference test, using samples with marginally elevated CA125 concentrations, defined as 35–200 U/mL (Supplementary Table 1). At 98% specificity, when applied to samples having ≥35 U/mL CA125, a sensitivity of 72% was reached, 4.5 times higher than the sensitivity reported using the conventional CA125 ELISA. The real cross-reactivity of healthy and benign samples in the CA125-STn-LFIA could not be assessed as they remained well below the established LoD of the assay. The selection of clinical samples for this study, marginally CA125-elevated EOC and endometriosis samples, was made to create a particularly clinically challenging situation. In a recent report[29] an STn1242-mAb-based CA125 glycovariant assay in microtiter wells showed substantially improved specificity also against several other benign conditions.

CA125 is well known to be secreted in a multitude of cancerous conditions other than EOC and is increasingly being investigated for diagnostic and prognostic potential. In a previous article[30], we sought to test two CA125 glycovariants (including the STn1242 antibody based) developed for EOC along with the conventional CA125 ELISA using metastatic breast cancer samples and benign controls. Interestingly, the findings showed that conventional CA125 assay was discriminating the controls from breast cancer cases significantly better (AUC = 0.884) than the CA125 glycovariant-based assays (AUC 0.572 for CA125-STn). These results are in line with the widely held notion that glycosylation is tissue specific[8].

The possibility of tissue and cancer type specificity would be a further valuable characteristic of glycovariants of conventional

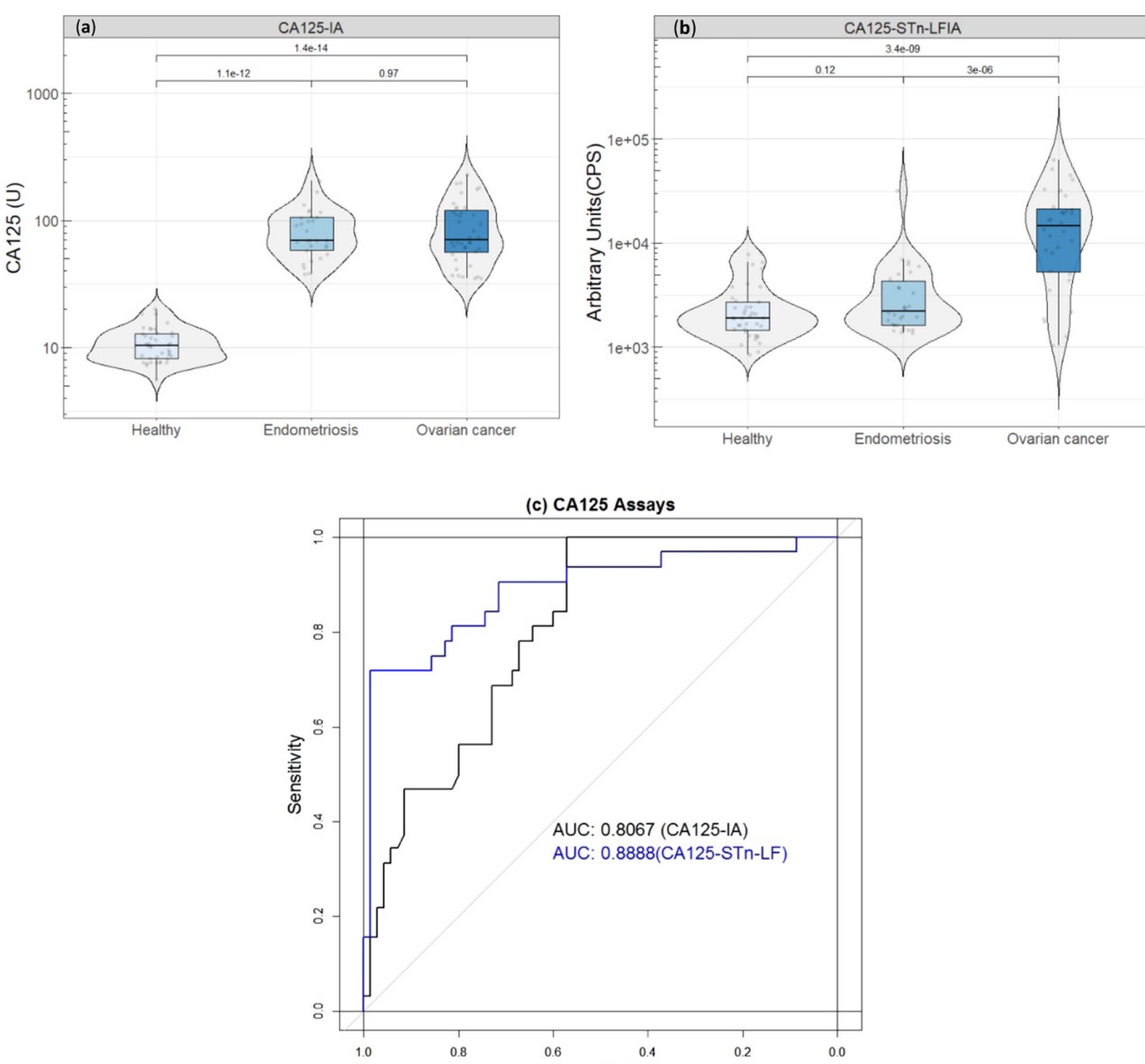

**Fig. 2 Evaluation of the developed assay.** We evaluated the assay using healthy controls ($n = 40$), benign endometriosis ($n = 30$), and EOC samples ($n = 32$). The healthy controls were <35 U/mL and the EOC and endometriosis samples were marginally elevated for serum CA125 concentrations (35–200 U/mL). **a** Discrimination of the EOC from benign endometriosis and healthy controls using the conventional CA125 test. **b** The developed CA125-STn-LF test significantly discriminated between the tested EOC from endometriosis and healthy samples. The line through the middle of the boxes corresponds to the median and the lower and the upper lines to the 25th and 75th percentile, respectively. **c** Receiver operating curves (ROC) describing the performance of CA125-STn-LFIA in comparison to the CA125-IA. The relatively higher AUC from CA125-STn-LFIA indicates high level of test accuracy and specificity towards CA125 of EOC cases. The 95% confidence intervals for CA125-STn-LFIA and CA125-IA were 0.8054–0.9576 and 0.7188–0.8826, respectively.

tumor markers in addition to the ability to discriminate the many confounding benign conditions. This aspect needs to be carefully explored and in future plans to evaluate CA125 in other cancers, e.g., lung, colorectal, and pancreatic cancer, it is imperative to start by screening for cancer type-specific glycoforms. The findings of the present proof-of-principle study call for independent confirmation using early-stage EOC samples, with different histological subtypes, as well as samples from normally confounding benign disease groups. Other common cancers known to express CA125 would also be tested for recognition, or lack thereof of the CA125-STn glycovariant.

The superiority of the developed LFIA test over conventional CA125 ELISA is linked to the specific and efficient capturing of the STn motifs of the cancerous isoforms of macromolecular MUC16/CA125 and the use of UCNP to provide the analytical sensitivity using a simple rapid immunochromatographic protocol. The high density of STn1242-mAb on the test line enables rapid, efficient capture of the STn-CA125 motif associated with EOC. The 4602 mab is M11-like and belongs to group B (ISOBM classification), recognizing a protein epitope of CA125 and showing no cross-reactivity with CA19-9, CA15-3, CYFRA21-1, CEA, AFP, and HE4[31]. The use of UCNPs as the 4602 mAb

carrier provides, through the bio-avidity effect, strong binding with the target analyte and allow measurement of minute concentrations of the antigen. As compared to the conventional CA125-IA or to the microtiter well immunoassays, the CA125-STn-LFIA is considered a rapid assay with 2 min hands-on time. However, a shorter read-out time could be feasible in future work. Regarding the technical aspects of the developed LF strip, we extensively screened different materials of each of the LF components (i.e., NC membrane, sample pad, and conjugate pad). Different criteria have been considered while evaluating each material. A number of NC membranes were studied in terms of flow kinetics, interference, and test line signals. The selected membrane showed consistent flow of the serum and other reaction liquids, minimum background signals and higher test line signals in comparison to the remaining membranes. The glass fiber pad was employed as a sample pad and as conjugate pad. The preblocked pad was additionally impregnated to modulate chemical variability of the sample while holding the detector particles as well. This aided to decrease the variability between different materials and facilitaes easier transfer of the reaction liquid to the NC membrane. The absorbent pad was selected based on the bed volume in order to accomodate the full volume of the reaction liquid while creating no backflow.

In future, we will conduct a technical validation and a large-scale clinical validation of the developed test. Following the large-scale validation, the developed LFIA could become a routine companion diagnostic tool for the early detection of EOC. The dramatically improved cancer specificity encourages us to apply the concept of lateral flow platform for non-invasive detection of cancers with poor survival rates such as pancreas and lung cancers.

## Methods

**Ethical statement**. Informed consent was obtained from all subjects. The Ethics Committees of the Hospital District of Southwest Finland and the University of Turku, Turku, Finland, approved the use of the clinical materials applied (ETMK 53/180/2009 for EOC and NCT01301885 for endometriosis and healthy controls).

**Clinical samples**. Endometriosis and ovarian cancer serum samples were run on the conventional CanAg® ELISA CA125 immunoassay, from Fujirebio, according to the manufacturer's instructions. The endometriosis samples that showed marginally elevated CA125-IA (35–200 U/mL) were specifically opted for this study. Thus, 30 endometriosis and 32 EOC serum samples were run using the CA125-STn-LFIA. Regarding the healthy women, 40 serum samples were tested (Supplementary Table 1). We have evaluated the same cohort earlier (healthy, endometriosis, and EOC) with the plate-based CA125-STn assay developed[21,29].

**Reagents, materials, and apparatus**. The CA125 isolated from NIHOVCAR-3 cell line (OVCAR3-CA125) was obtained from Fujirebio. As the nonmalignant source of the antigen, we isolated CA125 from pooled ascites fluid from liver cirrhosis patients (LC-CA125). Monoclonal mouse IgG antibodies 4602 mAb (Cat. No. 100598) and 4601-mAb (Cat. No. 628), against the peptide epitope of CA125 were provided by Oy Medix Biochemica. The STn1242-mAb (CAT. No. 119-01R) was provided by Fujirebio. The CA125 peptide targeting antibodies Ov197 (CAT. No. 203-01R) and Ov185 (CAT. No. 212-01R) were kindly provided by Fujirebio. Carboxyl-functionalized UCNPs (RD Upcon®540-L-C1-COOH; NaYF$_4$: Yb$^{3+}$, Er$^3$ nanoparticles of 55 × 43 nm with 15-nm coating were provided by Kaivogen OY (Turku, Finland). 1-ethyl-3- (3-dimethylaminopropyl)-carbodiimide (EDC) and sulfo-N-hydroxysuccinimide (NHS) were purchased from Thermo Fisher Scientific (USA). Bovine serum albumin (BSA) and 2-(N-morpholine) ethane sulfonic acid (MES, ultra-pure grade, 99.0%) were purchased from Sigma-Aldrich Co. (St. Louis, MO, USA). A hi-flow nitrocellulose (NC) membrane LFNC-C-BS023 (Nupore membranes, Ghaziabad, India), was the membrane of choice. A pre-blocked glass-fiber pad, with a grade 8951 (Ahlstrom-Munksjö, Finland), was additionally blocked using borate buffer pH 7.5, 0.05% BSA, and 20% tween-20 to be employed as the sample pad. The glass-fiber pad serves as a sample receiver, as well as conjugate pad. The cellulose fiber pad, CFSP223000 (Millipore, USA) was the absorbent pad of choice. Non-contact dispenser SciSpotter™ was purchased from Scienion AG (Berlin, Germany). CM 5000 Guillotine 5 Cutter was purchased from Bio-Dot (Switzerland). Upcon® strip reader was obtained from Labrox (Turku, Finland). The de-ionized water used to prepare the reagents and buffers was purified through the EVOQUA instrument (Processing Water, Finland).

**Principle of the CA125-STn-LFIA**. After loading a serum sample into the sample pad, the sample solution migrates through the strip towards the absorbent pad. If the sample contains EOC-associated-CA125, it will be sandwiched by the anti-STn-mAb and the 4602 mAb-UCNPs on the test line. If the sample contains no EOC-CA125 or below the LoD, the reporter only binds to the control line. The concentration of CA125 in each sample is correlated with intense of the peak measured by the Upcon®. The validity of the test is dependent on peaks of control line of each LF strip. If no peak on the control line, the LF test is considered invalid and the test should be repeated using a new LF strip.

**Preparation of the reporter**. The covalent conjugation between amino groups of the reporter antibody and the activated surface carboxyl groups of the UCNPs was performed according to the earlier described protocol with slight modifications[26]. Activation was performed using N-hydroxysulfosuccinimide (sulfo-NHS) and N-(3-dimethylaminopropyl)-N'-ethylcarbodiimide (EDC)-chemistries. Sulfo-NHS and EDC (Sigma) were incubated with 0.5 mg UCNPs in 50 mmol/L MES buffer (pH 6.1). The conjugation reaction contained 5 M NaCl, 500 mM MES buffer (pH 6.1) and 30 μg of the reporter antibody at room temperature (RT) for 30 min. Unoccupied surfaces of the UCNPs were blocked by incubating the bioconjugated reporter with 50 mM glycine (pH 11) at RT for 30 min. The storage buffer contained 25 mM borate (pH 7.8), 150 mM NaCl, 0.1% NaN$_3$, 2 mM KF, and 0.2% BSA[28].

**Selection of the LF materials**. Prior the selection of the LF materials (i.e., NC membrane, sample pad, and conjugate pad), we tested a number of nitrocellulose membranes and different candidate materials to be employed as sample pad and as absorbent pad. To analyze the NC membranes, a serum pool, collected from healthy women, was used as a blank. Moreover, the ovarian cancer cell line OVCAR3-CA125 was spiked into the serum pool as positive calibrator. The NC membranes were studied in terms of flow kinetics, interference, and test line signals. The selected membrane, LFNC-C-BS023, showed low interference to the immunoassay and rapid kinetics, leading to optimum sensitivity. The flow rate of the membrane was around 69 s/4 cm. Then, we tested different pre-blocked conjugate and sample pad materials in addition to length of these pads. The findings suggested that the 8951-sample pad (24 mm length) outperformed the remaining materials.

**Assembly of the CA125-STn-LFIA strips**. The LF membranes were assembled on a plastic support. The test and control lines on the LF strips were dispensed using the SciSpotter onto the nitrocellulose membrane LFNC-C-BS023. The test line antibody was dispensed at a distance of 10 mm from entry point of the NC membrane, with an optimized concentration of 0.5 mg/mL, into 10 mM Tris-HCl buffer at pH 8 and 5% (v/v) ethanol. The control line with a concentration of 0.4 mg/mL of rabbit-anti-mouse polyclonal antibody (code No. Z 0259), Dako Denmark AS (Glosturp, Denmark). was printed at a distance of 5 mm from the test line using the same printing buffer. After printing, the lines were dried at +35 °C for 1 h. Subsequently, the sample pad and the absorbent pad were laminated on the backing card with an overlap of 2 mm between the materials. Kenosha adhesive tape (Kenosha, Netherlands) was used to cover the overlapping membranes; to ensure that the membranes are correctly laminated. The assembled card was then cut into 4.8 mm wide lateral flow strips using the Guillitoin cutter. To dry the reporter particles on LF strips, the particles were diluted in a buffer composed of 50 mM Tris, pH 7.5, 500 mM NaCl, 2 mM KF, 7.5 mM CaCl$_2$, 0.04% NaN$_3$ (w/v), 0,05% Tween-20 (w/v), 1% BSA (w/v). The UCNPs bioconjugate were added manually to the prepared lateral flow strips. The particles were dried using a desiccator for 1 h at +35 °C. The reaction buffer consisted of 50 mM Tris pH 7.5, 0.05% Tween-20, 2 mM KF, 0.04% NaN$_3$, 0.05 mg/mL mIgG, 1% BSA. The wash buffer consisted of 10 mM Bis-Tris pH 6.5, 250 mM NaCl, 1% BSA, 0.5% Tween, 0.1% Germal II, 2 mM KF for the wash step.

**CA125-STn-LFIA protocol and evaluation**. First, 35 μL of a serum sample was pipetted onto each strip. Then, 25 μL of assay buffer. Then, 50 μL of wash buffer was added to each strip and left to run in room temperature for 30 min before measuring the signals using the the Upcon. In the Upcon, luminescence was measured at 540 nm upon excitation of the UCNPs at wavelength of 976 nm. The assay optimization was performed using spiked pooled serum from healthy women. The performance of CA125-STn-LFIA was evaluated in terms of sensitivity, specificity, and limit of detection (LoD). The LoD of the CA125-STn-LFIA was determined by detection of the spiked OVCAR3-CA125 in the pool of healthy women sera. A series of concentrations, 0.0, 1, 5, 10, 50, 100, 500, and 1000 U/mL, of the analyte were tested. The blank calibrator was analyzed in 50 replicates, the calibrators of 5, 10, and 50 U/mL were analyzed in 20 replicates and the calibrators of 1, 100, 500, and 1000 U/mL were analyzed in six replicates. The LoD of the CA125-STn-LFIA, which expresses the lowest possible concentration of analyte that can be detected with reasonable certainty. The LoD of the assay was determined from the standard curves and four parameters logistic regression of Origin 2016 was applied for fitting the standard curves. The LoD was calculated from the standard curve with a cutoff calculated as follows:

$$LoD = \mu_B + 1.645\sigma_B + 1.645\sigma_S$$

Where $\mu_B$ is the mean of blank calibrator measurements, $\sigma_B$ is the standard

deviation of blank measurements, and $\sigma_S$ is the standard deviation of low concentration calibrators' measurements. The sensitivity and specificity of the developed assay was estimated using R software. Accuracy and precision were indicated as recovery and CV%, respectively. Regarding the kinetics, the strips were repeatedly measured every 10 min upon adding the wash buffer.

**Statistics and reproducibility**. Origin 2016 (b9.3.2.303) was used to process the experimental data and the LoD calibration curve. We performed statistical analyses using the R software (http://www.r-project.org/), version 3.6.2. Box plots were done with Tidyverse (version 1.3.0)[32] and ggpubr (version 0.2.5) R packages[33]. The pROC R package was used for the The Receiver operating characteristics (ROC) analysis; ggplot R package for boxplot[34]. Within the tested samples group, the two assays were compared using the bootstrap test, provided in the pROC R package, for two correlated ROC curves. The measured CA125 concentrations of each assay were used as the classifier. We compared and evaluated the clinical performance of the developed CA125-STn-LF through ROC curves, computing areas under the curve (AUC). The $P$-values analyses were calculated using R. A $P$-value of <0.05 was considered significant in all statistical tests.

**Reporting summary**. Further information on research design is available in the Nature Research Reporting Summary linked to this article.

## Data availability

The data that support the findings of this study are available from the corresponding author upon reasonable request. Source data for Fig. 2 is available in Supplementary Data 1.

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

## Acknowledgements
We gratefully acknowledge Jenna Jacobino, Taina Heikkilä, Joonas Terävä, Parvez Syed, and Teppo Salminen at the Department of Biotechnology, University of Turku, Finland, for excellent technical assistance. This work was supported by the Jane and Aatos Erkko Foundation, Finland; the Nordic cancer Union, Denmark [grant number 194914].

## Author contributions
Conceptualization: S.B., K.G., and K.P.; investigation: S.B.; visualization: S.B. and J.L.; formal analysis: S.B., K.H., and J.L.; writing—original draft: S.B.; writing—review and editing: K.P., K.G., J.L., K.H., H.H., S.M.T., M.P., J.H., A.P., U.L.; resources: K.H., J.H., M.P., A.P., and K.P.; funding acquisition: K.P., K.G., and U.L.; supervision: K.P.

## Competing interests
The authors declare no competing interests.
