## [Peer Review File · Communications Biology]

Reviewers' comments:

Reviewer #1 (Remarks to the Author):

This manuscript reported the used of quantitative lateral flow immunoassay using upconverting nanoparticles (UPNPs) coupled with an optical reader for the detection of CA125 focus on epithelial ovarian cancer (EOC). The authors demonstrate using CA125-STn-LFIA for detection of CA125 and compared the studies with conventional CA125 immunoassay, with better sensitivities and detection limits. The studies is systematic and interesting, and the quality of the results are good. However, there are several major points need to be addressed. Below are my specific comments:

- 1) One of the concern is the real clinical application of CA125 for EOC, as CA125 itself is not specific for EOC, and it also founds in other cancer diseases. The authors could provide further discussion and explanation related to the implementation of CA125-STn-LFIA in real clinical practice for EOC.
- 2) Second, the introduction section may provide a more comprehensive overview on the existing biomarkers for EOC and highlight the clinical significant of CA125 for EOC.
- 3) Regarding the layout of the CA125-STn-LFIA in Fig 1a, the conjugate pad was placed at the end of the assembled membranes. It is unclear how the sample flow from the sample pad could interact with the UPNPs labelled antibodies, and then reaching to the nitrocellulose membrane for detection. There is no adsorption pad in the CA125-STn-LFIA, and how to drive the sample fluid flow? The sample flow rate also related to the supplementary Fig 3, which is unclear how the signal is being developed over the time.
- 4) The authors should explain in more details the different between the glycovariant-based approach and conventional approach, and highlight the technical advancement of the glycovariant-based approach over state-of-the-art.
- 5) What are the storage stability and shelf life of the CA125-STn-LFIA which is important to implement the LFIA for clinical use?

Reviewer #2 (Remarks to the Author):

This manuscript describes a quantitative lateral flow immunoassay utilizing the upconverting nanoparticles (UCNPs) for the detection of STn-glycosylated CA125 antigen to differentiate epithelial ovarian cancer (EOC) from benign endometriosis and healthy controls in patient-derived serum samples. The results look promising and the idea is useful and straightforward. However, the material part, lateral flow assay, and even STn-glycosylated CA125 as the biomarker possess limited novelty. Based on the above reasons, I don't recommend this manuscript to be accepted for publishing on Communications Biology.

Reviewer #3 (Remarks to the Author):

In this article, the authors present a quantitative lateral flow immunoassay (LFIA) of aberrantly glycosylated CA125, extremely relevant for diagnostically challenging samples with marginally elevated CA125. The novelty of their approach is the use of a monoclonal antibody against a sialyl-Tn antigen. To develop their LFIA they use commercial upconverting nanoparticles (UCNP) (from

Kaivogen, Upcon) and commercial strip reader developed to read fluorescence emission from these UCNP (from Labrox, Upcon). All the analytical work is sound and rigorously conducted.

Results are novel for researchers using LFIA tests, as they might not be familiar with the Upcon system, allowing a very interesting quantitative detection with high sensitivity. Innovation of this work is related to the use of an anti-sialyl-Tn antigen monoclonal antibody to selectively detect aberrantly glycosylated CA125.

The article should be published after some clarifications:

1) Information is lacking on the selection of the sample pad, type of nitrocellulose in the membrane and conjugate pad materials. Why were these particular materials selected? Were some other options tried?

2) Usually the LFIA test areas (sample pad, elution membrane, and conjugate pad) need some blocking (with albumin or casein, for instance) to improve assay specificity. Was any blocking tried?

3) The authors should clarify in the beginning of the article what they mean by a "tracer", as they use it along the article to identify different conjugates. Is it the same as "reporter", a name they use only once on line 250?

4) The authors mention (lines 91-92) "Three concentrations of the model analyte were spiked into serum pools of healthy individuals as calibrators and were compared to a blank calibrator. ". Two questions about this:

a) What is the "model analyte"? Is it the "sample" referred in the legend of Supp fig. 3? Why use a different name?

b) What is the content of the "blank calibrator"?

5) The authors continue (lines 93-94): "The optimum binding (i.e. highest S/N ratio) was reached between 30 to 40 min of incubation (supplementary Fig. 3).", but there is no S/N info in Supp fig. 3. Please explain this conclusion.

Reviewer #1 (Remarks to the Author):

This manuscript reported the used of quantitative lateral flow immunoassay using upconverting nanoparticles (UPNPs) coupled with an optical reader for the detection of CA125 focus on epithelial ovarian cancer (EOC). The authors demonstrate using CA125-STn-LFIA for detection of CA125 and compared the studies with conventional CA125 immunoassay, with better sensitivities and detection limits. The studies is systematic and interesting, and the quality of the results are good. However, there are several major points need to be addressed. Below are my specific comments:

- 1) One of the concern is the real clinical application of CA125 for EOC, as CA125 itself is not specific for EOC, and it also founds in other cancer diseases. The authors could provide further discussion and explanation related to the implementation of CA125-STn-LFIA in real clinical practice for EOC.*

Response to comment #1: We thank the referee for raising the discussion about the mentioned points and for the positive feedback.

We do agree with the reviewer in that CA125, using conventional immunoassay techniques, is known to be elevated in many other cancers besides ovarian cancer. However, it is also well established that glycosylation is frequently tissue specific.

In our previously published article (Gidwani K et al *Mol. Aspects of Medicine* 2019) we set out to test the CA125 glycovariants (initially developed for ovarian cancer) along with the conventional CA125 using the metastatic breast cancer samples and benign samples. Interestingly we found that the conventional CA125 assay discriminated the controls from breast cancer cases statistically significantly ($\alpha=0.05$) and much superior (AUC= 0.884) to the CA125 glycovariant assays (AUC= 0.572 for CA125-STn). These results are in line with the widely held notion that glycosylation is tissue specific. Consequently, with future plans to evaluate CA125 in other cancers, e.g. lung, and pancreatic cancer, it is imperative to start by identifying possible tissue and cancer specific glycovariants.

The envisioned clinical use is based on the decisively enhanced disease specificity both against the many benign conditions confounding the diagnostic evaluation and against other cancers (which as mentioned above call for extended evaluations). Combined with the UCNP assisted LFIA platform a future point-of care tools could allow rapid on-spot identification of patients with a highly increased risk for EOC to be directed to a definitive diagnostic work-up.

Changes made in response to comment #1: the discussion above has been added to the revised manuscript. In page 5, lines (186-201), we added the following sentences: “In our previously published article ³⁰, we sought to test the CA125 glycovariants developed for ovarian cancer along with the conventional CA125 using metastatic breast cancer samples and benign samples. Interestingly, we found that the conventional CA125 assay was discriminating the controls from breast cancer cases significantly ($\alpha=0.05$) better (AUC=0.884) than the CA125 glycovariant assays (AUCs being 0.572 for CA125-STn). These results are in line with the widely held notion that glycosylation is tissue specific ⁸. The possibility of tissue and cancer type specificity would be a further valuable characteristic of glycovariants of conventional tumor markers in addition to the ability to discriminate the many confounding benign conditions. This aspect needs to be carefully explored and in future plans to evaluate CA125 in other cancers, e.g. lung, colorectal, and pancreatic cancer, it is imperative to start by screening for cancer type specific glycoforms. The envisioned clinical use of the CA125-STn-

LFIA is based on the decisively enhanced disease specificity both against the many benign conditions confounding the diagnostic evaluation and against other cancers (which as mentioned above call for extended evaluations). Combined with the UCNPs assisted LFIA platform future point-of-care tools could allow rapid on-spot identification of patients with high risk for EOC to be directed to a definitive diagnostic work-up.”

2) *Second, the introduction section may provide a more comprehensive overview on the existing biomarkers for EOC and highlight the clinical significant of CA125 for EOC.*

Response to comment #2: we thank the referee for this suggestion, we provided this information in the introduction of the revised manuscript.

Changes made in response to comment #2: in page 1, lines 23-41, we added: “CA125 commonly also rises in several benign gynecologic conditions. In cases of clinical suspicion of EOC, there is a need for a simple POC test, that doctors could conduct at time of presentation. By measuring ovarian cancer associated CA125, the high-risk patients could be directed rapidly to a specialist; for definitive diagnostic work-up. A simple CA125 test of decisively increased EOC specificity is pivotal to reduce false positive results that expose patients to unnecessary invasive diagnostic procedures and to improve the treatment outcome and the survival rate of the patients ⁶.

Quantification of CA125 has been performed using conventional CA125 immunoassays (CA125IA). The CA125IA utilize the specificity of different monoclonal antibodies targeting protein-epitopes on CA125, including OC125 and M11, or OV197 like antibodies ⁷. These assays have been widely implemented for the monitoring of disease progression or regression, rather than for early detection of ovarian cancer. The inadequate specificity of CA125 impedes its use in early-stage EOC diagnosis and disease progression ^{5, 8, 9}. For this reason, supplementary biomarkers to CA125 such as HE4 ¹⁰ or multi-modal diagnostic tests (ROMA, ROCA, OVA1, and Overa) with CA125 as the key component have been studied ¹¹. More recently, several endeavors combined multiple tumor markers to increase clinical sensitivity and specificity of EOC detection, but CA125 has remained the preferred biomarker ^{12, 13}. The results of a large prospective ovarian cancer screening study (UKTOCS), combining ultrasound and serial CA125 measurements, showed that this screening strategy may improve early detection and reduce disease mortality ⁵.”

3) *Regarding the layout of the CA125-STn-LFIA in Fig 1a, the conjugate pad was placed at the end of the assembled membranes. It is unclear how the sample flow from the sample pad could interact with the UCNPs labelled antibodies, and then reaching to the nitrocellulose membrane for detection. There is no adsorption pad in the CA125-STn-LFIA, and how to drive the sample fluid flow? The sample flow rate also related to the supplementary Fig 3, which is unclear how the signal is being developed over the time.*

Response to comment #3: there was a writing mistake in the Fig 1a. The wrong labelling of the material made the liquid flow unclear. Regarding the signal development in supplementary Fig 3, the Y-axis shows a logarithmic scale of the measured test line signals. The signal response improves by time and stabilizing around 40 to 60 mins. There was a substantial increase in signals at longer incubation times. However, we were interested in working within the window of 30 mins read-out.

Changes made in response to comment #3: in page 3, line 111, the correct labeling is changed now to be “**absorbent pad**” instead of conjugate pad. The developed lateral flow strip has no separate conjugate pad. There is only one pad that acts as a sample receiving pad and as a conjugate pad.

4) *The authors should explain in more details the different between the glycovariant-based approach and conventional approach, and highlight the technical advancement of the glycovariant-based approach over state-of-the-art.*

Response to comment #4: the differences between glycovariant-based approach and the conventional approach are now included in the corrected manuscript.

Changes made in response to comment #4: in page 1, lines 30-34, we added the following sentences “Quantification of CA125 has been performed using conventional CA125 immunoassays (CA125IA). The CA125IA utilize the specificity of different monoclonal antibodies targeting protein-epitopes on CA125, including OC125 and M11, or OV197 like antibodies ⁷. These assays have been widely implemented for the monitoring of disease progression or regression, rather than for early detection of ovarian cancer”.

Regarding the advancement of glycovariant assays, in page 2, lines 51-57, the sentences explain the glycovariant approach “The aberrant glycosylation is conjoined with malignant transformation, tumor progression, and metastasis of many tumor types and leads to altered serum glycoprofiles ¹⁷. Notably, targeting aberrant elevated STn expression has attracted interest due to the improved clinical specificity and sensitivity for ovarian cancer detection ^{18, 19}. Therefore, glycan biomarkers of EOC could be a viable differential diagnostic tool. Gidwani et al. recently demonstrated novel CA125 glycovariant-based assay utilizing fluorescent-europium nanoparticles, which improves discrimination of EOC from benign endometriosis disease compared to conventional immunoassays ^{20, 21}.”

5) *What are the storage stability and shelf life of the CA125-STn-LFIA which is important to implement the LFIA for clinical use?*

Response to comment #5: we agree with the reviewer that shelf life studies are crucial for lateral flow tests. However, such studies are preferably conducted at production like facilities with full control of humidity and temperature. We have conducted such stability studies internally in our research laboratory, but the validity of these preliminary studies is doubtful

due to inadequate control of humidity and temperature. Therefore, the storage stability is not addressed in the present manuscript.

Reviewer #2 (Remarks to the Author):

This manuscript describes a quantitative lateral flow immunoassay utilizing the upconverting nanoparticles (UCNPs) for the detection of STn-glycosylated CA125 antigen to differentiate epithelial ovarian cancer (EOC) from benign endometriosis and healthy controls in patient-derived serum samples. The results look promising and the idea is useful and straightforward. However, the material part, lateral flow assay, and even STn-glycosylated CA125 as the biomarker possess limited novelty. Based on the above reasons, I don't recommend this manuscript to be accepted for publishing on Communications Biology.

Response to comment of reviewer #2: We thank the referee for raising a concern about our work. We respectfully disagree with the comment made by the referee regarding the limited novelty due to the following reasons. We claim that the novelty of our study is an aggregate of numerous factors to address a pressing clinical need. The use of nanoparticles (in this case fluorescence upconverting NP) as the antibody carrier enables, through the bio-avidity effect, the required analytical sensitivity while maintaining the cancer specificity inherent in the STn antibody. The UCNP technology in itself offers easy, low-cost quantitative detection using a well-established point-of-care platform.

Along with differential diagnostics in EOC and endometriosis with our glycovariant of CA125 assay compared to the conventional CA125 assay. The other novelty observed in our recent study (unpublished), the influence of conventional CA125 expressing in metastases from non-ovarian cancers (gastrointestinal, lung and neuroendocrine cancer), while our novel glycosylated CA125 assay was undetected. This further suggests an enhanced epithelial ovarian cancer specificity of CA125-STn. These results are in line with the widely held notion that glycosylation is tissue specific.

Reviewer #3 (Remarks to the Author):

In this article, the authors present a quantitative lateral flow immunoassay (LFIA) of aberrantly glycosylated CA125, extremely relevant for diagnostically challenging samples with marginally elevated CA125. The novelty of their approach is the use of a monoclonal antibody against a sialyl-Tn antigen. To develop their LFIA they use commercial upconverting nanoparticles (UCNP) (from Kaivogen, Upcon) and commercial strip reader developed to read fluorescence emission from these UCNPs (from Labrox, Upcon). All the analytical work is sound and rigorously conducted. Results are novel for researchers using LFIA tests, as they might not be familiar with the Upcon system, allowing a very interesting quantitative detection with high sensitivity. Innovation of this work is related to the use of an anti-sialyl-Tn antigen monoclonal antibody to selectively detect aberrantly glycosylated CA125.

The article should be published after some clarifications:

- 1) Information is lacking on the selection of the sample pad, type of nitrocellulose in the membrane and conjugate pad materials. Why were these particular materials selected? Were some other options tried?**

Response to comment #1: We thank the referee for raising the discussion about the mentioned points and for the positive feedback.

The developed lateral flow strip is composed of a sample pad, nitrocellulose membrane and an absorbent pad. A number of commercial nitrocellulose (NC) membranes were tested and compared. To analyze the membranes, a serum pool, collected from healthy women, was used as a blank. Moreover, the ovarian cancer cell line OVCAR3-CA125 was spiked into the serum pool as positive calibrator. The NC membranes were studied in terms of flow kinetics, interference, and test line signals. The selected membrane, LFNC-C-BS023, showed low interference to the immunoassay and rapid kinetics, leading to optimum sensitivity. The flow rate of the membrane was around 69 s/4cm. Then, we tested different pre-blocked conjugate and sample pad materials in addition to length of the pads. Then, the findings suggested that the G041 sample pad (24 mm length) outperformed the remaining materials.

Changes made in response to comment #1: in page 7, lines 267-277, we added the following paragraph in order to clarify the raised point: “**Selection of the LF materials.** Prior the selection of the LF materials (i.e NC membrane, sample pad, and conjugate pad), we tested a number of nitrocellulose membranes and different candidate materials to be employed as sample pad and as absorbent pad. To analyze the NC membranes, a serum pool, collected from healthy women, was used as a blank. Moreover, the ovarian cancer cell line OVCAR3-CA125 was spiked into the serum pool as positive calibrator. The NC membranes were studied in terms of flow kinetics, interference, and test line signals. The selected membrane, LFNC-C-BS023, showed low interference to the immunoassay and rapid kinetics, leading to optimum sensitivity. The flow rate of the membrane was around 69 s/4cm. Then, we tested different pre-blocked conjugate and sample pad materials in addition to length of these pads. Then, the findings suggested that the G041 sample pad (24 mm length) outperformed the remaining materials.”

2) *Usually the LFIA test areas (sample pad, elution membrane, and conjugate pad) need some blocking (with albumin or casein, for instance) to improve assay specificity. Was any blocking tried?*

Response to comment #2: the utilized membrane and materials are pre-blocked by the manufacturers. However, only, the sample pad was additionally blocked. The blocking reagents were selected upon different studies using different reagents and concentrations of each reagent. The optimum blocking reagents are as follows:

Sample pad: 10 mM borate buffer PH 7.5, 1 % bovine serum albumin (BSA), and 0.05 % tween 20.

Changes made in response to comment #2: in page 6, lines 237-242, we added the following sentences to explain this point: “A hi-flow nitrocellulose (NC) membrane LFNC-C-BS023 (Nupore membranes, Ghaziabad, India), was the membrane of choice. A pre-blocked glass-fiber pad, with a grade 8951 (Ahlstrom-Munksjö, Finland), was additionally blocked using borate buffer PH 7.5, 0.05 % BSA, and 20 % tween 20 to be employed as the sample pad. The glass-fiber pad serves as a sample receiver as well as conjugate pad. The cellulose fiber pad, CFSP223000 (Millipore, USA) was the absorbent pad of choice.”

3) *The authors should clarify in the beginning of the article what they mean by a “tracer”, as they use it along the article to identify different conjugates. Is it the same as “reporter”, a name they use only once on line 250?*

Response to comment #3: the word tracer was used to describe the conjugated label/reporter to an antibody and the word reporter was meant to refer to the unconjugated label/reporter particles.

Changes made in response to comment #3: through the whole text, the word tracer will be only used to describe the bioconjugates and the word reporter for unconjugated particles in order to unify the terminology and to be clear to readers. In page 2, lines 84-87, we added the following sentences: “An anti-CA125 mAb (4602, Oy Medix Biochemica, Finland), against the protein epitope was conjugated to the UCNPs to be used as a tracer (i.e reporter conjugated to antibody) (Fig. 1). The signals produced by the tracer corresponds to the intensity of signals measured from the test and control line.

4) *The authors mention (lines 91-92) “Three concentrations of the model analyte were spiked into serum pools of healthy individuals as calibrators and were compared to a blank calibrator. “. Two questions about this:*

a) What is the “model analyte”? Is it the “sample” referred in the legend of Supp fig. 3? Why use a different name?

Response to comment #4-a: the model analyte was purified CA125 from ovarian cancer cell line OVCAR3 (Fujirebio Diagnostics AB, Sweden). Yes, it is the same analyte spiked into healthy serum samples in order to study and evaluate the performance of the developed test.

Changes made in response to comment #4-a: in page 9, line 351, the word sample was replaced with “when blank calibrator or spiked calibrators (25, 50, and 100 U/mL) flow through”.

b) What is the content of the “blank calibrator”?

Response to comment #4-b: the blank calibrator was made of a pool of serum samples collected from healthy women.

Changes made in response to comment #4-b: we added the following sentence to describe the used blank in page 3, line 117-118, “and were compared to a blank calibrator (i.e non-spiked pool of serum samples collected from healthy women)”.

5) *The authors continue (lines 93-94): “The optimum binding (i.e. highest S/N ratio) was reached between 30 to 40 min of incubation (supplementary Fig. 3).”, but there is no S/N info in Supp fig. 3. Please explain this conclusion.*

Response to comment #5: the calibrator signals started to decrease around measurement time of 30 minutes and also. At the same time signals from the calibrators were also increasing, which will have a positive impact on calculations of S/N ratios.

Changes made in response to comment #5: in page 3, line (119), of the corrected manuscript, the phrase (i.e. highest S/N ratio) was removed from the manuscript to avoid confusion of readers.

REVIEWERS' COMMENTS:

Reviewer #1 (Remarks to the Author):

I am satisfied with the response and revision made by the authors. I would recommend the manuscript for publication.

Reviewer #3 (Remarks to the Author):

The authors answered appropriately all my questions and introduced the necessary clarifications to their manuscript. Their article is now fully consistent and interesting for the intended audience. The new version of the manuscript show be published as is.